# Risk Factors Associated with CIN2+ in Spanish Patients with L-SIL/ASCUS Cytology Collected from a Madrid Hospital

**DOI:** 10.3390/jpm12121944

**Published:** 2022-11-22

**Authors:** Virginia González González, Mar Ramírez Mena, Miguel Ángel Herráiz Martínez, Irene Serrano García, Pluvio J. Coronado Martín

**Affiliations:** 1Instituto de Salud de la Mujer, Hospital Clínico San Carlos, IdISSC, Facultad de Medicina, Universidad Complutense, 28040 Madrid, Spain; 2Research Methodological Support Unit, Hospital Clínico San Carlos, IdISSC, 28040 Madrid, Spain

**Keywords:** ASCUS, L-SIL, H-SIL/CIN2+, HPV, cervical cancer

## Abstract

The management of patients with L-SIL/ASCUS cytology is controversial and not clearly standardized. Objective: To analyze the risk factors associated with H-SIL/CIN2+ in a cohort of patients with ASCUS or L-SIL in a Pap smear. Methods: Between 2012 and 2022, 1259 eligible women with ASCUS/L-SIL were referred for colposcopy. The risk factors associated with H-SIL/CIN2+ were analyzed. The colposcopic study, conventional or assisted with dynamic spectral imaging (DSI), was performed in all cases. Guided biopsies were performed in cases of abnormal examination or random biopsies when no lesions were found. A LEEP was performed in H-SIL/CIN2+ results or persistent LSIL/CIN. Results: A normal or metaplastic specimen was found in 750 women (63.2%), LSIL/CIN1 in 346 (29.1%), and H-SIL/CIN2+ in 92 (7.7%). The presence of HR-HPV (OR = 2.1; IC 95% = 1.4–3.2), smoking habits (OR = 2.2; IC 95% = 1.4–3.5), and the performance of DSI combined with colposcopy (OR = 0.6; IC 95% = 0.37–0.83) were the factors associated with the detection of H-SIL/CIN2+. A summative effect of HR-HPV and smoking habit (OR = 2.9; IC 95% = 1.7–5.0) was observed in the detection of H-SIL/CIN2+. In multivariate analysis, the presence of HPV 16/18 was the unique independent factor associated with H-SIL/CIN2+. Conclusion: In women carrying an ASCUS/LSIL in the Pap smear, the unique independent factor predictive of H-SIL/CIN2+ is the presence of the HPV 16/18 genotype. Smoking women carrying ASCUS/LSIL with HR-HPV should be targeted for stricter follow-up to avoid an unsuspected H-SIL/CIN2+.

## 1. Introduction

Cervical cancer is considered the fourth most common malignant tumor in women around the world in terms of incidence and mortality [1].

Cervical cytology is an acceptable method for screening and is recommended in most European and American clinical guidelines. Nevertheless, its sensitivity for the detection of cervical intraepithelial neoplasia 2 or higher (H-SIL/CIN2+) is around 50%, not exceeding 80% in the best quality conditions [2]. The low sensitivity and reproducibility of the cytology have led to an analysis in recent years of the role of the HPV test in the secondary prevention of cervical cancer, either associated with cytology (co-test) or as an initial screening technique.

Women with abnormal cervical screening results or suspicious symptoms are usually assessed by conventional colposcopy. However, there is controversy regarding the proper evaluation and management of low-grade squamous intraepithelial lesions (LSIL) and atypical squamous cells of undetermined significance (ASCUS) in cervical cytologic diagnoses [3].

Colposcopy can evaluate the cervical lesions and detect the characteristics and type of precancerous or cancerous lesions. Indeed, it could help in orienting the biopsies of any area that seems abnormal. Furthermore, depending on the lesions’ extension, it could also help to determine which is the most appropriate treatment (cryotherapy, laser, cold conization, or the Loop Electrosurgical Excision Procedure [LEEP]) [3,4].

It is of the utmost importance to implement specific criteria for women with ASCUS/L-SIL, specifically for more cost-effective approaches to triaging women for colposcopy [5]. L-SIL is described in 2–3% of citologies and 12–16% of these will correspond with H-SIL/CIN2+ after colposcopy and biopsy [6]. The current Spanish Association of Cervical Pathology and Colposcopy (Asociación Española de Patología Cervical y Colposcopia—AEPCC) guidelines include the immediate referral to colposcopy in cases of L-SIL with HPV 16/18 genotype or when HPV was not performed. If a high-risk (HR) HPV genotype is detected with normal cytology, it considers the possibility of repeating the cytology in one year [6]. In cases of ASCUS, the prevalence of HPV ranges from 33 to 51%. In these cases, H-SIL/CIN2+ is found in 5–12% and cervical cancer between 0.1 and 0.2% [6]. In ASCUS, the management is less clear, taking into consideration the possibility of HPV determination, colposcopy, or repeating the cytology annually for two years. A complete anamnesis to provide a detailed report about the possible associated risk factors with H-SIL/CIN2+ in this type of patient could be an important aid in order to decide which management is more appropriate in each case, encouraging the application of more personalized medicine and avoiding the misdiagnosis of H-SIL/CIN2+. 

The aim of this populational study was to describe the features of a specific cohort of patients with a result of ASCUS/L-SIL in cervical cancer screening, and to analyze the risk factors associated with the detection of H-SIL/CIN2+.

## 2. Material and Methods

A transversal observational clinical study in a population of women with cervical screening results of ASCUS/L-SIL referred to colposcopy at Hospital Clínico San Carlos in Madrid, Spain, between 2012 and 2022 was performed. 

All women were managed following the guidelines of the Spanish Society of Cervical Pathology and Colposcopy (AEPCC) available at the time of the colposcopic assessment [7,8]. Cervical screening results of H-SIL, ASC-H, AGC, AIS, or cancer were excluded. Pregnant women, women with a history of pelvic radiation therapy, and women with a history of low genital tract cancer were not included. The DNA-HPV test was performed using the CLART^®^ HPV2 test, which detects and genotypes with high sensitivity and specificity 35 different types of HPV, including high-risk and low-risk types, in one assay. As a DNA control, it includes a set of primers to amplify a fragment of the human CFTR gene, while a spiked plasmid (1202 bp) is included as a PCR process control. The resulting visualized microarray is analyzed on a computer-guided reader with automated reading software. The genotypes 16, 18, 31, 33, 35, 39, 45, 51, 52, 56, 58, 59, 66, and 68 were considered HR-HPV. Clinical data and the personal backgrounds of each patient were recorded. 

Colposcopies with 3% acetic acid were performed on all women. The Schiller’s test was performed for cervical and vaginal evaluation [7,8]. A cohort of these patients was also studied with the assistance of the Dynamic Spectral Imaging System (DSI) [9,10]. Colposcopic findings were described according to Colposcopic Terminology of the International Federation for Cervical Pathology and Colposcopy (2011) [11,12]. Cervical punch biopsies were performed on all lesions detected. A random biopsy was performed in cases of a normal colposcopy assessment as part of the study design. An endocervical study was performed in cases of transformation zone (TZ) type 3. A LEEP was performed on women who had H-SIL/CIN2+ detected by punch biopsies, as well as cases of LSIL/CIN1 that had been present for at least two years. To classify the final histological result, in the event of a discrepancy between the punch biopsy and the LEEP specimen, the higher grade of either of the two results was considered. All colposcopies were performed by a unique colposcopist (PJC) in order to reduce interobserver variability. 

The sample size was calculated using a poblational reference of 2000 patients. Considering a poblational proportion of H-SIL/CIN2+ of 7%, with a CI of 95% and a 99% statistical power, the required sample size is 1112 patients.

### Statistical Analysis

Qualitative variables were summarized by their frequency distribution as well as quantitative variables by their mean and standard deviation (SD). The continuous, non-normally distributed variables were summarized by the median and interquartile range (IQR: P25–P75). In the case of qualitative variables, the chi-square test was used to assess the factors associated with H-SIL/CIN2+, or the Fisher’s exact test if more than 25% of the expected values were less than five. Quantitative variables were evaluated using the Student´s t-test. The magnitude of the association was evaluated using the odds ratio (OR) and its 95% confidence interval (CI). All statistical tests were two-sided. The statistical significance level was defined as *p* < 0.05. The computations were performed using IBM SPSS Statistics version 25 (Chicago, IL, USA) and Epidat 3.1 (Galicia, Spain). 

Different risk factors for cervical lesions were evaluated in the statistical analysis, considering the age, the presence of HPV, or other factors, as well as the number of sexual partners, smoking, or anal intercourse. A univariate analysis was performed to detect the factors associated with H-SIL/CIN2+. A multiple linear regression analysis was performed to identify the independent variables associated with H-SIL/CIN2+ among the variables with a *p*-value less than 0.10 in the univariate analysis, which were entered into the model using a forward/backward stepwise procedure.

The study was performed according to the STROBE guidelines and was approved by the Clinical Research Ethics Committee of Hospital Clínico San Carlos (C.I. 13/314-E). Signed informed consent was obtained from all participants.

## 3. Results

A population of 1219 patients was analyzed; 31 cases were excluded due to wrong or missing data. A final sample of 1188 women was evaluated for the study. Figure 1 describes the flow chart of patients included and excluded. In the final histology, a normal or metaplastic cervix was found in 749 women (63%), LSIL/CIN1 in 347 (29.3%), and H-SIL/CIN2+ in 92 (7.7%). A total of 54 out of 434 patients with HR-HPV (12.4%) presented H-SIL/CIN2+ in their final histology. Table 1 describes the main characteristics of the patient.

Table 2 shows the main variables and their role as risk factors for the presence of H-SIL/CIN2+ in the final histology. 

In the univariate analysis of the factors associated to H-SIL/CIN2+ there were found significant differences in the presence of HR-HPV (OR = 2.1; IC 95% = 1.4–3.2), smoking habit (OR = 2.2; IC 95% = 1.4–3.5), and the performance of the DSI map combined with conventional colposcopy (OR = 0.6; IC 95% = 0.37–0.83). 

In cases of HPV 16/18, the sensitivity to detect H-SIL/CIN2+ was 33.9% (IC 95% 24.3–43.6) with a specificity of 89.5% (IC 95% = 87.6–91.3). H-SIL/CIN2+ was diagnosed in 6.8% of the cases with no HPV 16/18, with a sensitivity to detect H-SIL/CIN2+ of 6.7% (IC 95% 3.6–9.8) and a specificity of 97.7% (IC 95% = 95.8–99.5). 

In the multivariate analysis, we included those factors considered individually relevant (*p* < 0.1) to be analyzed using a logistic regression model: HR-HPV, HVP 16/18, HR-HPV No. 16/18, anal intercourse, tobacco, and the DSI + CC study (Table 3). HPV 16/18 was the unique independent factor in the presence of unexpected H-SIL/CIN2+. 

## 4. Discussion

In patients with ASCUS/LSIL, it is of utmost importance to detect underlying H-SIL/CIN lesions (or immediate H-SIL/CIN2+) in order to avoid their progression to cancer. The detailed analysis of the potential risk factors associated with a H-SIL/CIN2+ could personalize the potential risk of being affected, improving patient experience, exposing fewer patients to diagnostic procedures, and lowering healthcare costs [13].

In women with HR-HPV, different cofactors have been described in the literature that increase the risk of cervical cancer and its precancerous lesions. Tobacco and HR-HPV infection, particularly by genotype 16/18 [13,14], are among them, as is the case in our series.

The immediate risk of H-SIL/CIN2+ in women with cytological L-SIL and positive HPV is 15% [15] and 7.7% in women with ASCUS [16]. In our series, 7.7% of the women carrying ASCUS/LSIL presented a histological confirmation of H-SIL/CIN2+, and 12.4% when they presented HR-HPV. 

Charlton et al. [13] found a significant increase in H-SIL/CIN2+ and adenocarcinoma in situ (AIS) in patients with a previous abnormal Pap test or biopsy [97.8% vs. 94.3%; OR = 2.44, 95% CI (1.03–5.76)] and a non-significantly higher rate in smokers (46.2% vs. 37.2%) and in women with ≥4 sexual partners reported (76.3% vs. 64.0%). However, once we put these into the regression model, the unique variable that was significantly associated with H-SIL/CIN2+ or AIS was having a prior history of an abnormal Pap or biopsy. 

As in our study, McIntyre-Seltman et al. [17] found a significant association between smoke (current smokers (OR, 1.7; 95% CI, 1.4–2.1) and past smokers (OR, 1.7; 95% CI, 1.2–2.4)) and the detection of H-SIL/CIN2+3+ in women with ASCUS/L-SIL. 

In our study, the presence of HR-HPV and current smoking habits were associated with an increase in H-SIL/CIN2+ detection. Furthermore, when analyzing together the presence of a smoking habit and the presence of HR-HPV, the detection of H-SIL/CIN2+ was higher than when considering both factors separately, which highlights the importance of insisting on stopping smoking in patients with HR-HPV. 

The fact that HPV 16/18 was the unique independent factor of the presence of occult H-SIL/CIN2+, HPV is the most recommended test for its detection. In patients with ASCUS/LSIL and the presence of HVP 16/18, the immediate risk of H-SIL/CIN3+ can reach 11%; on the contrary, genotypes different from 16 or 18 present an approximate risk of 3.7% [6,18]. With HPV screening and genotyping, a risk level stratification could be achieved, allowing the colposcopic examination and/or follow-up to be determined based on the associated risk factors. 

The performance of DSI added to the CC was found to be a protective factor (OR = 0.6; IC 95% = 0.37–0.83) in the detection of H-SIL/CIN2+. We did not find significant differences in other factors evaluated, such as HPV vaccination, postmenopausal status, or anal intercourse, although some of them have already been described as risk factors for developing cervical cancer (age or immunocompromise) [12].

The strength of this study is the large sample included, which allows for high precision in the results obtained. The patients have been evaluated by a unique and senior colposcopist, who followed the same protocols and criteria for every patient, which increased the feasibility of the study and reduced bias. A senior colposcopist’s performance of the colposcopy implies an increase in sensitivity compared to a junior [19], though this is not related to a specificity variation. Nevertheless, the unique operator may be considered a limitation of the study since there is not a second operator as recommended in the clinical trials. However, this is not possible in real clinical practice in an office, which is what this study tries to show. 

The factor associated with an unsuspected H-SIL/CIN2+ in women with ASCUS/LSIL in cervical cancer screening cytology was the presence of HPV 16/18 genotypes. Women carrying ASUS/LSIL with HR-HPV and who smoke should be targeted by meticulous medical evaluation, diagnostic testing, and follow-up in order to avoid an unsuspected H-SIL/CIN2+. 

## Figures and Tables

**Figure 1 jpm-12-01944-f001:**
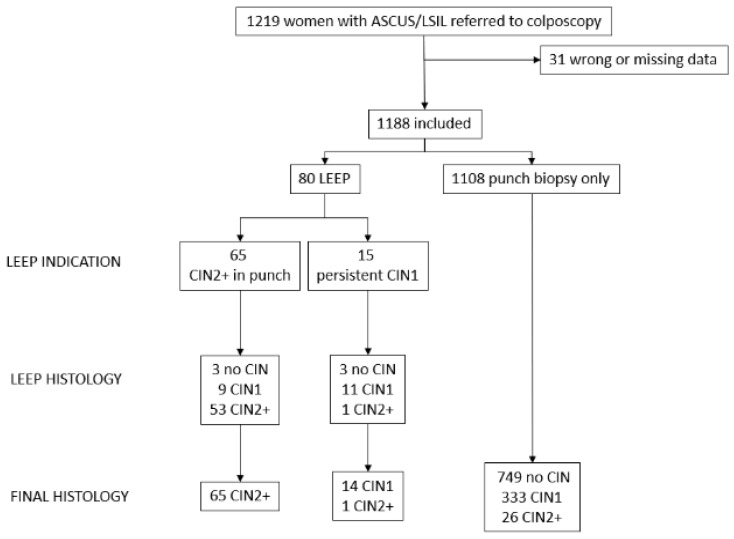
Flow chart of included and excluded patients and their anatomopathological results.

**Table 1 jpm-12-01944-t001:** Main characteristics of the cohorts.

	N (%)/Mean (SD) N = 1188
Age (years)	35.2 (10.4)
Age at first sexual intercourse	17.2 (5.3)
Number of sexual partners	7.3 (8.6)
Anal sexual intercourse	199 (16.8%)
Current Smoker	154 (12.9%)
Pap-smear	
ASCUS	472 (39.7%)
L-SIL	706 (59.4%)
CC/C-DSI result	
Normal/Metaplasia	458 (38.6%)
Changes grade 1	626 (52.7%)
Changes grade 2	93 (7.8%)
Cancer	1 (0.0%)
HPV	
Genotype 16/18	152 (12.8%)
High-risk HPV No. 16/18	282 (23.8%)
Low-risk HPV	124 (10.4%)
Negative	181 (15.2%)
Undetermined	439 (37.0%)
Inmunocompromise	
HIV	11 (0.9%)
Other	29 (2.4%)
Patients who required LEEP	90 (7.6%)
Normal/Changes grade 1	32 (2.7%)
Changes grade 2	58 (4.9%)

CC: Conventional colposcopy. C-DSI: CC assisted by the Dynamic Spectral Imaging System. SD: Standard deviation. ASCUS: Atypical squamous cells of undetermined significance. L-SIL: Low-grade squamous intraepithelial lesions. HPV: Human papillomavirus. HIV: human immunodeficiency virus.

**Table 2 jpm-12-01944-t002:** Univariate analysis of the features and their association with H-SIL/CIN2+ in the final histology.

	OR (IC 95%)	*p*
Age	0.8 (−1.1–2.7) *	0.4
HR-HPV	2.1 (1.4–3.2)	<0.001
HVP 16/18	12.7 (5.5–29.4)	<0.001
HR-HPV No. 16/18	3.1 /1.3–7.4)	0.01
≥5 sexual partners	1.4 (0.9–2.3)	0.12
Anal intercourse	1.6 (1–2.5)	0.08
Current smoker	2.2 (1.4–3.5)	<0.001
DSI + CC	0.6 (0.37–0.83)	<0.05
Smoking + HR-VPH	2.9 (1.7–5.0)	<0.001
HPV Vaccine	0.8 (0.4–1.5)	0.5
Postmenopausal status	0.4 (1.2–1.3)	0.1
Immunocompromise	1.8 (0.8–4.3)	0.2
HIV	1.9 (0.3–8.8)	0.33

High-risk HPV (HR-HPV No. 16/18) was defined as the presence of any of the genotypes 31, 33, 35, 39, 45, 51, 52, 56, 58, 59, 66, and 68. HIV: human immunodeficiency virus. * Mean differences (IC 95%).

**Table 3 jpm-12-01944-t003:** Logistic regression model of the relevant factors (*p* < 0.1) in the univariate analysis.

	OR (IC 95%)	*p*
HVP 16/18	12.7 (3.6–44.9)	<0.001
HR-HPV No. 16/18	3.1 (0.9–11.4)	0.08
Anal intercourse	0.5 (0.3–1.0)	0.08
Current smoker	0.2 (0.1–1.1)	0.06
DSI + CC	0.6 (0.4–0.9)	0.03
Smoking + HR-HPV	3.1 (0.6–17.1)	0.18

In the multivariate analysis, the relevant factors (*p* < 0.1) from the univariate analysis were included: HR-HPV, HVP 16/18, HR-HPV No. 16/18, anal intercourse, tobacco, and the DSI + CC study.

## Data Availability

The datasets generated and analyzed during the current study are available from the corresponding author upon reasonable request.

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
