# Peer review of "Risk Factors Associated with CIN2+ in Spanish Patients with L-SIL/ASCUS Cytology Collected from a Madrid Hospital"

_jpm, 2022, doi:10.3390/jpm12121944_

Round 1
Reviewer 1 Report
1. Please update the rank of cervical cancer in female. According to International Agency for research on cancer World health organization (https://gco.iarc.fr/today/home), cervical cancer is the fourth common cancer found in female in 2020.2. The detection rate of LSIL/ASCUS cytology in female population should be included in the introduction to increase the important of these group in cervical cancer screening.
3. Please describe more in the the CLART® HPV2 test protocol?
4. Line113, please correct the number of a normal was it 739 not 749?
5. Table 2, the second and third row, HPV16/18, HR-HPV No 16,18 and theirs OR to be the in same row.
6. The significance risk factors and OR presented in table 2 should be written in line 123-124 for example, the OR of HPV16,18 , 12.7 for CIN2+, so that, it can be compared to the next sentence "The presence of not detected H-SIL/CIN2+ was associated with the presence of HR- 128 HPV (OR=2.1; IC 95%=1.4-3.2; p<0.001)".
7. Apart form HR-HPV, smoking habit and DSI map written in Line 128-131, are there other risk factors with OR calculation of <CIN2 (Normal/HSIL) thus the reader can compared to calculated OR of HSIL/CIN2+ and see the important of HPV16,18 and smoking habit in the presence of CIN2+?
8. Is it possible to calculate the sensitivity of specificity of HPV16/18 pos, HPV16,18 neg vs HSI;/CIN2+ and <CIN2?
Author Response
Thank you for the opportunity to review the manuscript and for your comments and suggestions, which will help to improve the manuscript.
- Please update the rank of cervical cancer in female. According to International Agency for research on cancer World health organization (https://gco.iarc.fr/today/home), cervical cancer is the fourth common cancer found in female in 2020.
RE: Thank you for your review and comment, it has been updated.
- The detection rate of LSIL/ASCUS cytology in female population should be included in the introduction to increase the important of these group in cervical cancer screening.
RE: We agree with the reviewer. It has been included in the introduction section.
- Please describe more in the CLART® HPV2 test protocol?
RE: Included more details of the CLART technique in the material and methods section.
- Line113, please correct the number of a normal was it 739 not 749?
RE: thank you for your comment and for identifying this mistake. The correct number was in was placed in figure 1, the number in the main text was corrected.
- Table 2, the second and third row, HPV16/18, HR-HPV No 16,18 and theirs OR to be the in same row.
RE: Thank you. It has been changed.
- The significance risk factors and OR presented in table 2 should be written in line 123-124 for example, the OR of HPV16,18 , 12.7 for CIN2+, so that, it can be compared to the next sentence "The presence of not detected H-SIL/CIN2+ was associated with the presence of HR- 128 HPV (OR=2.1; IC 95%=1.4-3.2; p<0.001)".
RE: Table 2 has been modified to include the results
- Apart form HR-HPV, smoking habit and DSI map written in Line 128-131, are there other risk factors with OR calculation of <CIN2 (Normal/HSIL) thus the reader can compared to calculated OR of HSIL/CIN2+ and see the important of HPV16,18 and smoking habit in the presence of CIN2+?
RE: It is a good suggestion, but other risk factors associated with CIN2+ have not been calculated. Nevertheless, this comparison would be considered in future investigations.
- Is it possible to calculate the sensitivity of specificity of HPV16/18 pos, HPV16,18 neg vs HSI;/CIN2+ and <CIN2?
RE: It has been calculated and added in lines 145-148.
Reviewer 2 Report
The manuscript “Risk factors associated with CIN2+ in patients with L-SIL/ASCUS cytology” by González et al., aims to analyze the risk factor associated with the detection of HSIL/CIN2+ in patients with ASCUS/LSIL in cervical cancer screening. The authors reported that the presence of HPV16/18 genotypes is the main factor associated with ASCUS/LSIL. Additionally, they reported that smoke showed some significant association with HSIL/CIN2+. Despite a large number of collected patients (1219), the author's findings and conclusions were already observed and reported in many other published studies and didn’t bring any new concepts or information to the field. I would suggest that the authors re-write this study as a short communication focusing on their observation as a local population epidemiology report.
Author Response
Response to Reviewer 2 Comments
Thank you for the opportunity to review the manuscript and for your comments and suggestions, which will help to improve the manuscript.
The manuscript “Risk factors associated with CIN2+ in patients with L-SIL/ASCUS cytology” by González et al., aims to analyze the risk factor associated with the detection of HSIL/CIN2+ in patients with ASCUS/LSIL in cervical cancer screening. The authors reported that the presence of HPV16/18 genotypes is the main factor associated with ASCUS/LSIL. Additionally, they reported that smoke showed some significant association with HSIL/CIN2+. Despite a large number of collected patients (1219), the author's findings and conclusions were already observed and reported in many other published studies and didn’t bring any new concepts or information to the field. I would suggest that the authors re-write this study as a short communication focusing on their observation as a local population epidemiology report.
RE: Thank you for your review and comment, we consider that this study reinforces the knowledge of the risk factors associated with HSIL/CIN2+ in patients with L-SIL/ASCUS citology. We think that the importance of this study is that reflects real-world results, so it measures the effectiveness (and not the efficacy) of the colposcopy. In addition, this large sample has been performed by a unique colposcopist, which reduces the variability. In this way, this study gives to the scientific community a real vision of the medical practice in the management of cervical cancer screening. That is why we think these findings are worth to be considered for publication.
Reviewer 3 Report
This article is well constructed and clear, I added some comments directly in the PDF to make the readers understand the methodology better.

Author Response
Response to Reviewer 3 Comments
Thank you for the opportunity to review the manuscript and for your comments and suggestions, which will help to improve the manuscript.
It is necessary to add a sentence of backround.
RE: It has been included.
If we have de 95%IC there is no need for the p.
RE: If it is not inconvenient, we believe that including the p may help many readers to understand the statistical calculations and to see the dimension of the statistical significance. We have removed them from the text but we have maintained it in the tables.
A sample size part is missing in the section method, if it is a convenience sample, it must be said and explained.
RE: It has been included in the m&m section.
Cette partie ethique doit aller la fin de la mthode.
RE: It has been changed.
For the multivariate analysis it is necessary to indicate which type of test and statistical method has been used
RE: It has been included in the m&m section.
Add that this is the results of the univariate analysis
RE: It has been included.
Ad at the bottom of the table the adjusted variables on which the multivariate analysis has been done
RE: It has been included.
It lacks a clearly defined and developed limit of the study
RE: It has been modified.
Put in lower case.
RE: It has been modified.
Round 2
Reviewer 2 Report
Despite the manuscript modifications provided by the authors, it still does not bring any new concepts or information to the HPV/Cervical Cancer field. The authors` data report epidemiology findings in a specific population that may be different if the study was performed in a different population. I will suggest again modifying the writing to be clear on the manuscript that this is a populational study. The indication that it is a populational study should be also indicated in the manuscript title. Example: “Risk factors associated with CIN2+ in Spanish patients with L-SIL/ASCUS cytology collected from a Madrid hospital”.
Author Response
Thank you for your comments. The manuscript has been modified according to the suggestions. I hope you find it suitable for being published.
Reviewer 3 Report
thanks to the authors for their corrections and additions following my comments, I think their article is publishable in this form.
Author Response
Dear reviewer, thank you for your comments
Round 3
Reviewer 2 Report
The manuscript does not bring any new concepts or information to the HPV/Cervical Cancer field. However, after modifications, now it is clear that the manuscript is a descriptional report from a local Spanish population with L-SIL/ASCUS cytology diagnostic.